# Risk Factors for Excessive Social Media Use Differ from Those of Gambling and Gaming in Finnish Youth

**DOI:** 10.3390/ijerph19042406

**Published:** 2022-02-19

**Authors:** Sari Castrén, Terhi Mustonen, Krista Hylkilä, Niko Männikkö, Maria Kääriäinen, Kirsimarja Raitasalo

**Affiliations:** 1Health and Well-Being Promotion Unit, Finnish Institute for Health and Welfare, P.O. Box 30, 00271 Helsinki, Finland; kirsimarja.raitasalo@thl.fi; 2Social Sciences Department of Psychology and Speech-Language Pathology Turku, University of Turku, 20014 Turku, Finland; 3Department of Medicine, University of Helsinki, 00014 Helsinki, Finland; 4Sosped Foundation, 00510 Helsinki, Finland; terhi.mustonen@sosped.fi; 5Research Unit of Nursing Science and Health Management, University of Oulu, 90014 Oulu, Finland; krista.hylkila@oulu.fi (K.H.); mannikkon@gmail.com (N.M.); maria.kaariainen@oulu.fi (M.K.); 6School of Health and Social Care, Oulu University of Applied Sciences, 90101 Oulu, Finland; 7Medical Research Center Oulu, Oulu University Hospital, 90014 Oulu, Finland; 8Department of Social Sciences, University of Eastern Finland, 70210 Kuopio, Finland

**Keywords:** adolescents, ESPAD, excessive social media use, gaming, gambling, substance use

## Abstract

Purpose: Adolescents’ excessive social media use has characteristics similar to other addictive behaviours. This study aims to explore whether the same risk factors are associated with excessive social media use as with excessive gaming and gambling among Finnish adolescents. Methods: Multinomial logistic regression analyses were carried out using the European School Survey Project on Alcohol and Other Drugs data, collected from Finnish adolescents aged 15–16 in 2019 (*n* = 4595). Results: Excessive use of social media was more common among girls (reported by 46% of respondents) than boys of the same age (28%), whereas boys reported both excessive gaming (23%) and gambling (6%) more often than girls (4% and 1%, respectively). All differences between genders were statistically significant (*p* < 0.0001). Daily smoking was associated with a high risk of excessive gambling (AOR = 3.23) and low risk of excessive gaming (AOR = 0.27) but had no significant effect on excessive social media use. Cannabis use in the past 12 months was positively associated only with excessive gambling (AOR = 2.39), while past 12 months alcohol consumption increased the risk for excessive social media use (AOR = 1.25). Conclusions: Adolescent girls are at greater risk of excessive social media use than boys, while boys are at greater risk of excessive gaming and gambling. The associations with known risk factors are somewhat different for excessive use of social media as compared to excessive gambling and gaming and should be acknowledged when developing preventive measures for adolescents.

## 1. Introduction

It is a well-established fact that adolescents, especially boys, often participate in a variety of risky behaviours, such as the use of alcohol, cannabis, tobacco or nicotine products, or gambling and gaming [1,2,3,4]. These behaviours may demonstrate features of typical addictive behaviours like constant preoccupation, loss of control, high intensity and frequency, and continuation or escalation of the behaviour despite the occurrence of negative consequences and cause significant distress and impairment to an individual’s overall functioning [5]. These behaviours are often interwoven and may feed each other [6,7]. Due to the rapid development of technology, it has been suggested that excessive use of social media may have characteristics similar to excessive (or addiction-like) behaviours associated with gambling and gaming, thus potentially posing a risk for important areas of functioning among adolescents [8]. Therefore, it is essential to investigate possible factors that may be associated with excessive social media use to gain an up-to-date understanding of these behaviours among adolescents with the intention of developing effective preventive means.

Social networking sites, messaging apps, and other social media channels provide various types of activities and interactions, such as socializing, relaxing, sharing pictures, videos and stories, and chatting [9]. Browsing the web using smartphone applications has become an everyday activity, and the use of the internet for our daily errands is here to stay.

Even if social media use may be linked to positive psychosocial outcomes, such as an increase in social capital and connectedness [10,11], growing evidence has recently raised concerns about excessive use of social media and its associations with the user’s overall well-being: substance use, depression, anxiety, and other mental health issues like low self-esteem [11,12,13].

### 1.1. Excessive Use of Social Media

Excessive use of social media can be broadly defined as being excessively concerned about social media, being driven by a strong motivation to log on to or use social media, and devoting so much time and effort to social media use that it impairs other social activities, studies/job, interpersonal relationships, and/or psychological health and well-being [5]. These features are similar to other known excessive behaviours (gambling and gaming). Thus, a deeper understanding of their similarities and differences may increase knowledge of this relatively novel research area. Excessive gaming has been found to augment the symptoms of excessive engagement in social networking sites (SNS) and vice versa [14]. Current research has shown that excessive types of social media use are more common among women than men [15,16].

The prevalence of the excessive use of social media has been estimated to vary between 2.8% and 47% [16,17]. The variation of these estimations is due to differences in methodology and cultural reasons and the lack of consensus in defining excessive or problematic social media use and social media addiction as its utmost form [18]. In this study, the expression ‘excessive use of social media’ refers to self-perceived behaviour [19].

### 1.2. Excessive Gaming and Gambling

Excessive gaming and gambling have more established definitions compared to excessive social media use as more research has been conducted related to these behaviours. When referring to excessive gaming, the phrase ‘problematic gaming’ is often used [20]. Also, when referring to excessive gambling, the phrase ‘problem gambling’ is common. Both problem behaviors have established diagnostic criteria, Gambling Disorder, in the American Psychiatric Association’s [21] Diagnostic and Statistical Manual of Mental Disorders (5th ed.; DSM-5) and more recently in the World Health Organization’s 11th edition, ICD-11. The ICD-11 definitions for Gambling Disorder and Gaming Disorder resemble each other with the characterizations: impaired control over behaviour (e.g., onset, frequency, duration, termination, context); increasing priority given to the behaviour that causes negative consequences to the person’s life; continuation of behaviour despite the negative consequences [22]. Those who participate in gaming activities have a higher risk of engaging in gambling and vice versa [23]. The prevalence of problematic gaming among adolescents varies from 1.2% to 18.4%, depending on the measure used; the participation rate in gaming activities has been reported as being 59% among European adolescents in 2019 [2] (ESPAD Group 2020). Problem gambling prevalence rates for adolescents vary from 1.6% to 6.7% [24], 22% of European adolescents had gambled during the past year in 2019. Boys typically gamble and game more than girls and are more at risk of developing problems with gambling [4,25]. In this paper, we use the expression ‘excessive gaming and gambling’ throughout since we measured self-perceived excessive behaviours [26].

### 1.3. Risk and Protective Factors for Excessive Behaviours

Substance use has been linked to gambling and gaming in many studies [6,27]. There is also clear evidence that adolescents whose parents are aware of their whereabouts and activities are less likely to participate in risky behaviours [28,29,30]. Along with the influence of family and parents, the role of peers is developmentally important in adolescents’ lives, and peers can boost or prevent adolescents’ risky behaviours [31]. Also, factors related to school, for example, low school grades, poor school performance, and skipping school are often linked with risky behaviours [32], and these predisposing factors may also affect orientation to future plans. In contrast, participation in extra-curricular or leisure activities may protect against risky behaviours [33]. Other proximal risk factors, such as family situation/status (e.g., single-parent/two-parent household, parental separation) have also been linked to risky behaviours in youth [34].

As social media has become an important part of daily activities, especially among adolescents, and it is known that involvement in one risky behaviour may predict involvement in other risky behaviours [22], it is important to investigate whether excessive social media use relates to excessive gaming and gambling to find effective preventive measures. This study aims at exploring whether the same risk and protective factors are associated with excessive social media use and excessive gaming and gambling among Finnish adolescents (15- and 16-year-olds).

## 2. Materials and Methods

### 2.1. Data

The data for this study was gathered as part of the European School Survey Project on Alcohol and Other Drugs (ESPAD) in Finland in spring 2019. The target population was defined as (1) regular students who (2) turned sixteen in the calendar year of the survey and (3) were present in class on the day of the survey administration. This definition includes students who were enrolled in regular or general studies and excludes both special schools for students with learning disorders or severe physical disabilities and students who did not speak Finnish or Swedish (Finland’s national languages).

A two-stage systematic probability-proportional-to-size sampling method using the Nomenclature of Territorial Units for Statistics 2 regions (NUTS 2) as strata and schools as clusters (NUTS 2, 2020) was used to collect a nationally representative sample with a self-administered pen-and-paper questionnaire taken in the classroom. Participation was voluntary, and anonymity was ensured. Those outside the target group and those who had responded inconsistently or had clearly exaggerated or had left unanswered over half of the questions were removed from the data (*n* = 435). An analytical sample including only individuals with valid answers to all variables of interest was created, and the final data included 2027 male and 2133 female students. The school participation rate and the student response rate were both about 90%. The data were weighted to reflect a representative sample of Finnish ninth-graders [2,35].

### 2.2. Measures

The dependent variables: self-perceived excessive social media use, self-perceived excessive gaming, self-perceived excessive gambling, and independent variables: daily cigarette smoking, alcohol use, cannabis use, weekly sports, weekly hanging out with friends, future plans, family type, and parents’ education—together with their definitions and distributions—are presented in Appendix A.

### 2.3. Statistical Analysis

Cross-tabulation with Rao-Scott’s chi-square tests were applied to study the differences between boys and girls in excessive social media use, gaming, and gambling in the past 12 months in different categories of the independent variables.Multinominal logistic regression models were fitted separately for the outcome variables (excessive social media use, gaming, and gambling). The effects of each independent variable were first looked at separately. After that, all variables of interest were controlled for. Odds ratios with 95% confidence intervals were then derived from the full model as marginal estimates.

The sample design was taken into account in all analyses by adjusting the student-level standard errors for clustering effects. This was carried out using the survey procedures offered by SAS EG 7.1 (SAS/STAT^®^, 2011) [36] using NUTS2 regions (NUTS2, 2020) as strata and schools as clusters.

## 3. Results

Table 1 shows that 28% of 15–16-year-old boys and 46% of girls reported excessive social media use. To continue, 23% of boys and 4% of girls reported excessive gaming. The respective proportions for gambling were 6% and 1%. All differences between genders were statistically significant at the <0.001 level regardless of the background factors.

As the differences in all dependent variables (excessive social media use, gaming, and gambling) according to all independent variables were of the same direction for boys and girls, we decided to adjust the models for gender and not study boys and girls separately.

Table 2 shows that after all adjustments, girls had a higher risk for excessive social media use than boys (AOR = 2.18, 95% Cl 1.84–2.58) and, respectively, a lower risk for excessive gaming (AOR = 0.13, 95% Cl 0.09–0.17) and excessive gambling (AOR = 0.08, 95% Cl 0.04–0.16). Daily smoking was negatively associated with gaming (AOR = 0.38, 95% Cl 0.17–0.85) but had no significant effect on excessive social media use or gambling. In contrast, past 12 months alcohol use was positively associated only with excessive social media use (AOR = 1.25, 95% Cl 1.02–1.52) and past 12 months cannabis use was positively associated only with excessive gambling (AOR = 2.39, 95% Cl 1.22–4.71). Daily sports activity decreased the risk for gambling (AOR = 0.39, 95% Cl 0.19–0.80) but was not related to excessive social media use or gaming. Hanging around with friends was negatively associated with excessive gaming (AOR = 0.54, 95% Cl 0.40–0.72) but positively associated with excessive gambling (AOR = 3.34, 95% 1.77–6.32). Moreover, parental control was negatively associated with excessive gambling (AOR = 0.39, 95% 0.12–0.71) but had no significant effect on excessive social media use or gaming. However, aiming at high school and living in a nuclear family increased the risk for excessive social media use (AOR = 1.27, 95% Cl 1.05–1.54 and AOR = 1.28, 95% Cl 1.06–1.54, respectively). Parents’ education was not associated with any of the outcome measures.

## 4. Discussion

In the present study using a large sample of Finnish adolescents, we examined whether the same factors that are associated with other excessive behaviours (gambling and gaming) also apply to the rising phenomenon of excessive social media use.

In our data, the prevalence of self-assessed excessive social media use was 28% among boys and 46% among girls, which was higher than in previous studies [8,37,38]. The prevalence rate in the current study is not fully comparable to these previous studies due to different study populations and methodologies.

Girls reported excessive social media use more often than boys, which is in line with previous studies. Girls use web-based applications for social communication [39] and are involved in social networking sites and messaging more than boys [40], and girls also use social media for social activities more than boys [41].

Andreassen and colleagues [38] suggest that girls rather than boys are at risk of developing excessive behaviours toward activities that include social interaction. Peris et al. [42] found that girls feel pressured to present themselves with idealized attractiveness by editing their physical appearance before posting it on social media. This could lead to constant waiting for external approval, thus increasing the risk of developing social media addiction [43]. Earlier studies have also shown that girls use different kinds of social media platforms more than boys and more frequently. However, boys use discussion boards more than girls [44].

Spending hours on social media sites possibly fills the need of belonging, which is explained by the Fear of Missing Out (FOMO) [45]. Nowadays, especially during the COVID-19 pandemic, which has increased the time spent on social media [37,46] and related excessive behavioural patterns [37], social media serves as a space for adolescents where they get connected, add and discharge contacts, exchange experiences and information, and thus form their social identity with peers [47].

As expected, the prevalence of excessive gaming and gambling was much higher among boys than girls. Our results are in line with previous studies, demonstrating that boys are more interested in both gambling and gaming and are thus more vulnerable to developing problems with these behaviours [4,16]. This may be explained by the observation that boys tend to develop problems towards asocial or solitary activities (e.g., gaming) [38]. Boys are more likely to use video games as a primary means of interacting with friends, maintaining friendships, and enhancing social status, while girls use social media for peer communication [40]. Our results also suggest that girls are gaming more than gambling (4% vs. 1%). One explanation may be that, even if the availability of mobile play and other digital payment services have lured more women into the gambling sector, it is possible that they are less likely to take a risky approach to gaming.

The co-occurrence of behavioural and substance addictions has been noted [6]. Also, having a behavioural addiction increases the probability of developing other behavioural addictions [48]. Risk factors related to gambling in the present data were daily smoking and cannabis use, which have been previously found to be associated with problem gambling [6]. These findings emphasise the fact that when planning preventive programs and screening adolescents for possible excessive gambling behaviour, it is important to be aware of the possibility of co-occurring substance use and plan support or treatment accordingly.

Moreover, our results show that hanging out with peers is a much stronger predictor of excessive gambling activity than excessive gaming or social media use. Gambling behaviour possibly differs from gaming and use of social media behaviours. A possible explanation is that gambling can take place online and offline in land-based locations where real-life interaction takes place, and thus it is more strongly affected by peer influence, whereas amongst gamers and social media users, the interaction is mainly virtual [4].

Factors that seemed to protect from excessive gambling were engagement in daily sports activity and parental control, while these factors had no effect on excessive gaming and use of social media. This may indicate that, although these behaviours share commonalities, they do have differences. This notion is even more evident when exploring risk factors for excessive use of social media where the use of alcohol was a risk factor, as noted before [44], along with other factors such as high educational aspirations and living in a nuclear family. Family structure has also been reported to have an important role in adolescent risky behaviours. Growing up in a nuclear family is typically a protective factor for risky behaviours [49], yet in our data, the opposite was true regarding excessive social media use.

All in all, even though excessive gambling and gaming resemble excessive use of social media, excessive social media users seem to differ from gamblers and, to some extent, from gamers. Pontes [14] states that the detrimental effects on health and well-being are more pronounced among excessive gamers than among excessive social media users. As youth gambling is clearly associated with substance use, gamblers share similar characteristics with substance users more than gamers do [50]. Critically, our findings suggest that the group most at risk of developing excessive social media use consists of girls who typically are not considered being at risk for addictive behaviours [51]. Yet, further studies are needed to confirm this and identify individuals who might be more vulnerable to excessive use of social media.

### 4.1. Practical Implications

Both gambling and gaming disorders are noted as being public health issues [52,53] that cause various types of health and social harms to individuals. When planning prevention programs for excessive gambling and gaming behaviours, it would be beneficial to link with components of other similar behaviours, such as excessive social media use, to increase public awareness about the phenomena. As adolescents are at risk for all these behaviours, prevention programmes should be targeted at them. In practice, asking about the use of social media and possibly using a screening tool (for example, in health checks at schools, e.g., Bergen Social Media Addiction Scale [54]) along with screening other risky behaviours would provide information and help to those at risk.

### 4.2. Strengths and Limitations

This study has been conducted on a large sample with a high response rate, making the data representative of Finnish 15- to 16-year-olds. The survey captures almost the whole age group. Still, as always with self-reported data, there is a risk that students consciously or unconsciously do not give accurate, honest answers about their actual behaviours. They may be over-reporting as well as under-reporting, depending on what is socially desired/accepted in different contexts. However, a validity report on ESPAD [55] shows that only a very small minority (1–2%) do not answer questions on substance use honestly. There is no reason to assume that this wouldn’t be the case with questions on social media use, gaming, and gambling. Because the survey is cross-sectional, no conclusions about causality between the dependent and independent variables can be drawn. Another possible limitation is that the respondents’ self-perceived problems in social media (excessive use) were assessed with a few targeted questions and not with a full standardized measure (e.g., Bergen Social Media Addiction Scale).

## 5. Conclusions

This study found clear gender differences in adolescents’ risky behaviours. Adolescent girls seem to be at greater risk for excessive social media use compared to boys. Boys, in turn, are at higher risk for excessive gaming and gambling. In order to find effective preventive means for adolescents, early detection (screening) of various possible risky behaviours should be put in practice and take gender specific factors into account when developing preventive measures and providing support for those in need.

## Figures and Tables

**Table 1 ijerph-19-02406-t001:** Distributions of excessive social media use, excessive gaming, and excessive gambling according to the independent variables among Finnish 15–16 years old boys and girls in 2019 with Rao-Schott’s Chi-square test results on statistical significance.

		Excessive Social Media Use			Excessive Gaming			Excessive Gambling		
		Boys	Girls	*p* (*chisq*)	Boys	Girls	*p* (*chisq*)	Boys	Girls	*p* (*chisq*)
Total		28.4%	46.4%	0.001	22.7%	3.6%	0.001	6.4%	0.5%	0.001
**Independent variables**										
Daily smoking	Yes	16.1%	47.3%	0.001	6.7%	1.9%	0.001	19.4%	0.9%	0.001
	No	29.2%	46.4%	0.001	23.6%	3.7%	ns	5.6%	0.5%	0.001
Past 12 months alcohol use	Yes	26.9%	50.2%	0.001	19.6%	3.1%	0.001	8.8%	0.8%	0.001
	No	30.8%	40.7%	0.001	27.3%	4.2%	0.001	2.8%	0.2%	0.001
Past 12 months cannabis use	Yes	21.6%	40.1%	0.004	15.3%	1.5%	0.001	17.9%	2.0%	0.001
	No	29.4%	46.9%	0.001	24.0%	3.7%	0.001	4.9%	0.3%	0.001
Sports	Yes	28.1%	46.8%	0.001	22.4%	3.5%	0.001	5.6%	0.5%	0.001
	No	29.3%	44.4%	0.018	26.7%	6.1%	0.001	15.8%	0.6%	0.001
Hanging around with friends	Yes	25.3%	50.8%	0.001	17.6%	2.2%	0.001	9.7%	0.5%	0.001
	No	32.1%	41.7%	0.001	29.0%	5.0%	0.001	2.5%	0.6%	0.001
Parental control	Yes	28.7%	46.0%	0.001	22.3%	3.8%	0.001	5.0%	0.3%	0.001
	No	23.2%	49.4%	0.001	21.7%	2.3%	0.001	18.0%	2.7%	0.001
Aims at high school	Yes	31.4%	46%	0.001	24.3%	3.2%	0.001	4.9%	0.3%	0.001
	No	24.5%	47.4%	0.001	20.6%	4.9%	0.001	8.3%	1.3%	0.001
Parents education	Yes	27.8%	46.1%	0.001	20.7%	3.5%	0.001	7.3%	0.4%	0.001
	No	29.8%	47.0%	0.001	27.5%	4.2%	0.001	3.3%	1.1%	0.045
Nuclear family	Yes	30.7%	47.9%	0.001	23.1%	3.5%	0.001	6.1%	0.6%	0.001
	No	22.5%	43.6%	0.001	20.6%	3.7%	0.001	6.7%	0.4%	0.001

**Table 2 ijerph-19-02406-t002:** Risk factors for excessive social media use, excessive gaming, and excessive gambling among Finnish 15–16 years old boys and girls in 2019, unadjusted (OR) and adjusted (AOR) odds ratios with 95% confidence levels (CL).

		Social Media		Gaming		Gambling	
		OR (95% CL)	AOR (95% CL)	OR (95% CL)	AOR (95% CL)	OR (95% CL)	AOR (95% CL)
Gender (ref = boy)		2.18 (1.86–2.56)	2.18 (1.84–2.58)	0.13 (0.10–0.17)	0.13 (0.09–0.17)	0.08 (0.04–0.16)	0.08 (0.04–0.16)
Daily smoking (ref = no)		0.82 (0.58–1.17)	0.83 (0.58–1.21)	0.27 (0.13–0.53)	0.38 (0.17–0.85)	3.23 (1.61–6.47)	0.98 (0.43–2.21)
Past 12 months alcohol use (ref = no)		1.13 (0.95–1.34)	1.25 (1.02–1.52)	0.69 (0.53–0.88)	0.89 (0.65–1.20)	3.35 (1.89–5.94)	1.69 (0.99–2.89)
Past 12 months cannabis use (ref = no)		0.66 (0.49–0.91)	0.77 (0.57–1.06)	0.67 (0.44–1.05)	0.71 (0.40–1.25)	4.94 (2.72–8.95)	2.39 (1.22–4.71)
Sports (ref = no)		1.08 (0.76–1.56)	0.97 (0.68–1.38)	0.67 (0.40–1.14)	0.89 (0.54–1.49)	0.30 (0.14–0.65)	0.39 (0.19–0.80)
Hanging around with friends (ref = no)		1.02 (0.88–1.19)	1.06 (0.91–1.25)	0.58 (0.46–0.73)	0.54 (0.40–0.72)	3.66 (2.16–6.21)	3.34 (1.77–6.32)
Parental control (ref = no)		1.02 (0.79–1.33)	0.94 (0.70–1.26)	1.14 (0.68–1.90)	0.66 (0.39–1.13)	0.24 (0.14–0.42)	0.39 (0.21–0.71)
Aims at high school (ref = no)		1.49 (1.25–1.77)	1.27 (1.05–1.54)	0.79 (0.64–0.97)	1.03 (0.82–1.29)	0.37 (0.22–0.64)	0.71 (0.41–1.21)
Parents education (ref = no)		0.96 (0.80–1.16)	0.87 (0.71–1.06)	0.70 (0.51–0.97)	0.75 (0.53–1.06)	1.75 (0.95–3.24)	1.90 (0.98–3.69)
Nuclear family (ref = no)		1.23 (1.02–1.47)	1.28 (1.06–1.54)	1.26 (0.99–1.59)	1.10 (0.85–1.43)	1.08 (0.63–1.86)	1.35 (0.76–2.39)

## Data Availability

The ESPAD trend data are archived at the Italian National Research Council (CNR), and data can be used for research purposes (http://espad.org/databases, accessed on 4 May 2021).

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
