# Peer review of "Risk Factors for Excessive Social Media Use Differ from Those of Gambling and Gaming in Finnish Youth"

_ijerph, 2022, doi:10.3390/ijerph19042406_

Round 1

Reviewer 1 Report

The aim of the reviewed manuscript is interesting, however, the quality of the article leaves much to be desired (rather a poor introduction and review of the literature on the subject) and the results do not add a lot to the existing literature. 

Detailed comments and suggestions:

Title

-The title of the article sounds like a conclusion or not ending expression, it would be better to think about re-editing it and to change into more consistent with the study project as you compared excessive social media use and gambling/gaming risk factors, and there are many kinds of addictive behaviors, and as you studied not only risk factors but also protective factors e.g. weekly sports 

Abstract

-line 19,31 – why do the words “Adolescents’ /Adolescent girls start from a capital letter?

Introduction

-lines 80-81 there is a few spaces and a lack of brackets in this sentence, and the stylistic error e.g. using “when referring” twice time right after should be rephrased

-line 85-6 what means “21”? something is missing here, and too many spaces between words are used

-in paragraph 1.1 you started the description from the definition of excessive social media, and then was the prevalence of this phenomenon. In my opinion, a similar layout of the argument should be in paragraph 1.2. In addition, you should precisely characterize the difference between gaming and gambling and the symptoms of these two behavioral addictions

-line 86-87 the end of the sentence “(…)developing problems” – what kind of problems?

-the subtitle 1.3. sounds stylistically incorrect, I suggest rephrasing it. In addition, this paragraph is poorly described and should be better organized, especially in the context of 

study aims; the title suggested that you had described risk factors of all addictive behaviors, while you only mentioned about few (categories social, educational, and substance abuse – we even do not know what kind of substance you were thinking about) in the context of gambling/gaming – this should be corrected

-line 103-104 I do not think that school performance, low school grades, etc. are social determinants;

- line 116 – two spaces and dots

-add the study hypothesis with justification, why you have chosen to measure these variables e.g.: daily cigarette smoking, alcohol use, cannabis use, weekly sports, weekly hanging out with friends, future plans, family type, and parents’ education. Moreover, some of these factors are not risk factors but protective factors – this must be corrected (especially in the context of the title)

Materials and Methods

-add short information about the participants e.g. M, SD of age, the gender distribution

-add the description of all scales used in the study and its psychometric properties, you only listed the studied variables without any information 

-why do you name independent variables (risk factors) as outcomes?

Results

-you used two names for dependent variables: background characteristics in the methods and materials section, and background factors, in my opinion, both are misleading, your factors are independent variables, and the dependent variables are outcomes as you are tried to explain the risk of developing addiction by using independent variables.

-Table 1 

– add the title for the first column;

- what kind of results are in the first row it has no name in the first column?  

-add a note to explain symbol p(chisq) 

-line 169, you used in the sentence: for both and girls”, shouldn’t be for boys and girls?

Table 2

- you used 95% cl, and the confidence level should be from the capital letters

-add a note to explain all symbols used in the table

Discussion

-Line 197 – add brackets for the number of referenced literature

  - line 212-13, the construction of the sentence suggests that FOMO is one of the psychological needs, while it is not – must be corrected

-why did you introduce FOMO in the discussion section, you should be paid more attention to your results 

-Line 235 – please delete the reference “(Sussman et al., 2017)” and leave the number in accordance with the journal’s guidelines

- Line 276 – what means 16? – correct the way of citation

- line 184 – what means 54?

-line 188 – what kind of covariates did you use? There was not a word about it in the manuscript methodological section

Conclusions

- the conclusions are rather poor and should be re-editing

I do not agree that increased awareness of risk factors related to excessive social medial use” is a practical implication

Author Response

Responses to the reviewers’ comments on the manuscript

ID:IJERPH- 1587569

Title: Risk Factors for excessive social media use differ from those of other addictive behaviors

Dear Editor and Reviewers,

We genuinely appreciate the effort and care that you went through with our paper. The reviews comments were thorough and offered us advise on how to improve our paper. With these comments in mind, our paper has gone revision. Below, we describe and then give detailed step-by-step responses to each reviewer comment and revisions are marked with track-change – function in the manuscript.

Rev I:

The aim of the reviewed manuscript is interesting, however, the quality of the article leaves much to be desired (rather a poor introduction and review of the literature on the subject) and the results do not add a lot to the existing literature. 

Response: We thank the reviewer for this point of view, which can be well understandable with the old title and aims. We have modified the title, the structure of introduction:  (subheadings 1.2. and  1.3.) as well as the aims. We hope that these revisions are now suitable and more clear. We have checked the review of the literature and feel that it is up – to- dated.

Detailed comments and suggestions:

Title

-The title of the article sounds like a conclusion or not ending expression, it would be better to think about re-editing it and to change into more consistent with the study project as you compared excessive social media use and gambling/gaming risk factors, and there are many kinds of addictive behaviors, and as you studied not only risk factors but also protective factors e.g. weekly sports 

Response: The title has been changed: “Risk factors for excessive social media use differ from those of gambling and gaming in Finnish youth”.

Response: We thank the reviewer pointing this out and protective factors is now added to the subheading 1.3 and added to the aims.

Abstract

-line 19,31 – why do the words “Adolescents’ /Adolescent girls start from a capital letter?

Response: these typing errors are corrected.

Introduction

-lines 80-81 there is a few spaces and a lack of brackets in this sentence, and the stylistic error e.g. using “when referring” twice time right after should be rephrased

Response: This is corrected.

-line 85-6 what means “21”? something is missing here, and too many spaces between words are used

Response: This is corrected.

-in paragraph 1.1 you started the description from the definition of excessive social media, and then was the prevalence of this phenomenon. In my opinion, a similar layout of the argument should be in paragraph 1.2. In addition, you should precisely characterize the difference between gaming and gambling and the symptoms of these two behavioral addictions

Response: The paragraph of 1.2 is rephrased using the same logic and presenting order than 1.1.

-line 86-87 the end of the sentence “(…)developing problems” – what kind of problems?

Response: This is clarified as following:  …developing problems with gambling.

-the subtitle 1.3. sounds stylistically incorrect, I suggest rephrasing it. In addition, this paragraph is poorly described and should be better organized, especially in the context of 

Response: Subtitle is corrected and the paragraph is re-organized.

study aims; the title suggested that you had described risk factors of all addictive behaviors, while you only mentioned about few (categories social, educational, and substance abuse – we even do not know what kind of substance you were thinking about) in the context of gambling/gaming – this should be corrected

Response: We thank the reviewer for pointing out these discrepancies. We have amended the aims and title based on this good notion and hope that is now clearer.

-line 103-104 I do not think that school performance, low school grades, etc. are social determinants;

Response: Yes, we agree, good point. This expression has been changed into “factors related to school”.

- line 116 – two spaces and dots

Response: Corrected.

-add the study hypothesis with justification, why you have chosen to measure these variables e.g.: daily cigarette smoking, alcohol use, cannabis use, weekly sports, weekly hanging out with friends, future plans, family type, and parents’ education. Moreover, some of these factors are not risk factors but protective factors – this must be corrected (especially in the context of the title)

Response: We prefer to express research questions rather than hypothesis. The justification of choosing the used background variables is presented in Subheading 1.3. We have re-phrased the aims and the title.

Materials and Methods

-add short information about the participants e.g. M, SD of age, the gender distribution

Response: All participants were of same age (15-16 years), only the yer of birth has been asked. Those who were not born in 2003 have been excluded from the analytical sample. Therefore, adding descriptive statistics on age was not necessary. Please see the first paragraph (2.1.) description of the target population (2). Gender distribution is added as requested.

-add the description of all scales used in the study and its psychometric properties, you only listed the studied variables without any information 

Response: We have provided all requested information in our previous version in the Appendix and hope that reviewers will have access to this material when reviewing our responses.

-why do you name independent variables (risk factors) as outcomes?

Response: The use of terms have been corrected throughout the manuscript so that we speak about independent and dependent variables

Results

-you used two names for dependent variables: background characteristics in the methods and materials section, and background factors, in my opinion, both are misleading, your factors are independent variables, and the dependent variables are outcomes as you are tried to explain the risk of developing addiction by using independent variables.

Response: See response to the previous comment.

-Table 1 

– add the title for the first column;

Response: Please see the modified Table 1.

- what kind of results are in the first row it has no name in the first column?  

Response: Please see the modified Table 1.

-add a note to explain symbol p(chisq) 

Response: Please see the modified Table 1.

-line 169, you used in the sentence: for both and girls”, shouldn’t be for boys and girls?

Response: Thank you, this has now been corrected.

Table 2

- you used 95% cl, and the confidence level should be from the capital letters

Response: This has now been corrected.

-add a note to explain all symbols used in the table

Response: All requested corrections are completed.

Discussion

-Line 197 – add brackets for the number of referenced literature

Response: This has now been corrected.

  - line 212-13, the construction of the sentence suggests that FOMO is one of the psychological needs, while it is not – must be corrected

-why did you introduce FOMO in the discussion section, you should be paid more attention to your results 

Response: We thank the reviewer for this notion, thus the sentence is rephrased and a new concept of FOMO is only mentioned briefly.

-Line 235 – please delete the reference “(Sussman et al., 2017)” and leave the number in accordance with the journal’s guidelines

Response: This has now been corrected.

- Line 276 – what means 16? – correct the way of citation

Response: Thank you for pointing this out, this was incorrect reference, which is not corrected and a new reference is added to the references with ref nro 54.

- line 184 – what means 54?

Response: We could not find this error in the MS.

-line 188 – what kind of covariates did you use? There was not a word about it in the manuscript methodological section

Response: This issue has now been corrected and suggested wording of independent and dependent variables are now used.

Conclusions

- the conclusions are rather poor and should be re-editing

I do not agree that increased awareness of risk factors related to excessive social medial use” is a practical implication

Response: The conclusions have now been rephrased.

Please find the response letter attached where the other Rev (II) comments and responses for your information. 

Reviewer 2 Report

The manuscript entitled "Risk factors for excessive social media use differ from those of other addictive behaviours in youth" is an important work that describes the results of the ESPAD study among a representative sample of Finish adolescents. The authors compared the tendency to addictive behaviors, including excessive use of social media, gaming, and gambling, regards some demographic variables and substance use (i.e., smoking, alcohol, and cannabis use). Although the research is not innovative, the risky behaviors change continuously due to environmental changes related to civilization development. Examining risky behaviors patterns is crucial to preparing efficient prevention programs in schools. Although the manuscript is well-written, some changes may improve it substantially.

Methods

  • The method section needs complementation. Could you add detailed information about the number and percent of people excluded for each reason (i.e., outside the target group, inconsistent responses, missing data, etc.)?
  • Procedure should be described in a separate section. Please move study design and performance information to this new section (from the Participants section). It should also be added information on how the authors deal with missing data.
  • Each variable mentioned in the Measures section should be implemented by describing the scale of the answer or categories of response if it is more appropriate. Also, the coding system (i.e., how the particular categories of answers were coded into numbers for the logistic analysis purpose) should be presented in this section. It is necessary to understand the results of this study.
  • Supplementary Table is mentioned in the manuscript (page 3, line 140), but not provided. However, this information about the distribution of demographic variables should be presented in the manuscript, as a characteristic of the sample, in the Participants section (instead of in the supplement).

Results

  • More statistics are necessary to report associations between gender and excessive social media, gaming, and gambling. The number of people in each group for comparison (n) exact chi-square statistic (Χ2) with df, as well as effect size (e.g., φ or Cramer’s V), should be added in Table 1 besides percentage and p-value. See APA guidelines for more details about how to report statistical results.

Author Response

Rev II

The manuscript entitled "Risk factors for excessive social media use differ from those of other addictive behaviours in youth" is an important work that describes the results of the ESPAD study among a representative sample of Finish adolescents. The authors compared the tendency to addictive behaviors, including excessive use of social media, gaming, and gambling, regards some demographic variables and substance use (i.e., smoking, alcohol, and cannabis use). Although the research is not innovative, the risky behaviors change continuously due to environmental changes related to civilization development. Examining risky behaviors patterns is crucial to preparing efficient prevention programs in schools. Although the manuscript is well-written, some changes may improve it substantially.

Methods

The method section needs complementation. Could you add detailed information about the number and percent of people excluded for each reason (i.e., outside the target group, inconsistent responses, missing data, etc.)?

Response: The Finnish ESPAD data is part of the larger ESPAD data set, thus cleaning of the whole ESPAD data is centrally performed in Italy and this type of information has never been requested before in relation to other publications. We do know that in Finland 46 students did not participate in the study (no parental consent provided). In practice, 34 cases were students of one particular school in Helsinki, where non- returned parental consent – form was interpreted as no consent to participate to the study. The rest of the cases (N=12) were students from various schools, whose parents did not give their consent to their child. In addition 31 students did not want to participate to the study. We feel that this information is not necessary to report in this MS.

Please see addition (N=435) in the manuscript line 136.

Procedure should be described in a separate section. Please move study design and performance information to this new section (from the Participants section). It should also be added information on how the authors deal with missing data.

Response: We checked the style of the IJERPH and modified this section accordingly  and rephrased the first subheading 2.1. Data as instructed in the IJERPH.

Handling missing data is described as follows in the MS: An analytical sample including only individuals with valid answers to all variables of interest was created, and the final data included 2027 male and 2133 female students.

Each variable mentioned in the Measures section should be implemented by describing the scale of the answer or categories of response if it is more appropriate. Also, the coding system (i.e., how the particular categories of answers were coded into numbers for the logistic analysis purpose) should be presented in this section. It is necessary to understand the results of this study.

Response: We have provided all requested information in our previous version in the Appendix and hope that the reviewers will have access to this material when reviewing our responses.

Supplementary Table is mentioned in the manuscript (page 3, line 140), but not provided. However, this information about the distribution of demographic variables should be presented in the manuscript, as a characteristic of the sample, in the Participants section (instead of in the supplement).

Response: We hope that the reviewers have now access to the Appendix, which was provided earlier. Regarding the request of characteristics of the participants: in this study all the participants were of same age and the number of girls and boys is now added (please see line 138).

Results

More statistics are necessary to report associations between gender and excessive social media, gaming, and gambling. The number of people in each group for comparison (n) exact chi-square statistic (Χ2) with df, as well as effect size (e.g., φ or Cramer’s V), should be added in Table 1 besides percentage and p-value. See APA guidelines for more details about how to report statistical results.

Response: We thank the reviewer for this opinion. However, we disagree with the reviewer regarding this issue. Presenting Cramer’s Vs in case of chi-square tests is often biased in case of large samples and does not add anything to the results. In regards to df’s, this should be fine, since all variables are dichotomous (df=1). Please see the title of Table 1 however with some re-phrasing.

Please find the full response letter attached, where you cab see the rev I comments and our responses for your information. 

Round 2

Reviewer 1 Report

The changes made to the manuscript are sufficient and it can be published.